# A Near-Infrared CMOS Silicon Avalanche Photodetector with Ultra-Low Temperature Coefficient of Breakdown Voltage

**DOI:** 10.3390/mi13010047

**Published:** 2021-12-29

**Authors:** Daoqun Liu, Tingting Li, Bo Tang, Peng Zhang, Wenwu Wang, Manwen Liu, Zhihua Li

**Affiliations:** 1Institute of Microelectronics, Chinese Academy of Sciences, Beijing 100029, China; liudaoqun@ime.ac.cn (D.L.); litingting@ime.ac.cn (T.L.); tangbo@ime.ac.cn (B.T.); zhangpeng1@ime.ac.cn (P.Z.); wangwenwu@ime.ac.cn (W.W.); 2School of Electronic Electrical and Communication Engineering, University of Chinese Academy of Sciences, Beijing 100049, China

**Keywords:** impact ionization, silicon avalanche photodetector, CMOS technology, punch-through structure, temperature stability, near-infrared photodetector

## Abstract

Silicon avalanche photodetector (APD) plays a very important role in near-infrared light detection due to its linear controllable gain and attractive manufacturing cost. In this paper, a silicon APD with punch-through structure is designed and fabricated by standard 0.5 μm complementary metal oxide semiconductor (CMOS) technology. The proposed structure eliminates the requirements for wafer-thinning and the double-side metallization process by most commercial Si APD products. The fabricated device shows very low level dark current of several tens Picoamperes and ultra-high multiplication gain of ~4600 at near-infrared wavelength. The ultra-low extracted temperature coefficient of the breakdown voltage is 0.077 V/K. The high performance provides a promising solution for near-infrared weak light detection.

## 1. Introduction

Avalanche photodetectors (APDs) have been widely used for weak light detection due to their large internal avalanche gain. Specifically, APDs play key roles in light detection and ranging (LiDAR) systems to covert the weak echo optical signal into the electrical one [1,2,3,4,5]. The following requirements for APDs should be satisfied to enable the capability of weak light detection in LiDAR systems [4]. First, APDs with high-enough gain are required to ensure long detectable distance. Moreover, APDs with high temperature stability are preferable due to their high tolerance to ambient temperature variation. Finally, the cost of such APDs should be reasonable to promote widespread application. Thus, exploring APDs with high gain, high temperature stability, and low cost is desirable. 

For LiDAR applications, the light detection can be implemented by p-i-n photodetectors [6,7], linear-mode APDs [2,8,9], and Geiger-mode APDs such as single-photon avalanche photodetector (SPAD) [1,3,10], and multipixel photon counters (MPPC) or silicon photomultipliers (SiPM) [4,11]. Photodetectors with internal multiplication gain are preferable due to their higher sensitivity in weak light detection over p-i-n detectors. Linear-mode APD with a simpler device structure is also preferred because of the comparable time resolution to Geiger-mode APD [4]. Recently, Ge/Si APD [2,12] and APDs based on III-V semiconductor materials [13,14] with ultralow temperature dependency of the breakdown voltage have been demonstrated. However, these APDs have much higher multiplication excess noise than that of Si APDs due to the lower k-value (the ratio of ionization coefficient of holes over electrons) of silicon. Most importantly, the Si APDs have lower cost due to their compatibility with the complementary metal oxide semiconductor (CMOS) infrastructure. Although some commercial Si APDs are available in the market, a dedicated fabrication process is required [15,16,17]. Additionally, these APDs generally have much high temperature coefficients of breakdown voltage. Such high temperature coefficient puts high demand on the automatic gain control circuit. Although a high-performance Si APD with lower temperature coefficient of 0.32 V/K was demonstrated [8], the wafer thinning process and the double-sided metallization required by such devices are not available for 8-inch or 12-inch standard CMOS foundries.

In this work, a silicon avalanche with high performance is elaborately designed and fabricated in an 8-inch pilot line using 0.5 μm CMOS technology. The dark current, photocurrent, and capacitance were measured based on the fabricated device. The proposed Si APD shows excellent performance, including low-level dark current, ultra-low temperature coefficient of breakdown voltage, strong photoresponse at near-infrared wavelength, and low-level capacitance. The responsivity and current gain are extracted from the above measurement results. The Si APD shows extremely high responsivity and consequently current gain near the breakdown point. It is worth noting that the temperature coefficient of the Si APD is 0.077 V/K, which can greatly improve the design freedom of the driver-circuit. The high performance provides a promising solution for weak light detection in LiDAR systems.

## 2. Device Design and Fabrication

Figure 1a shows the cross-section schematic of the fabricated Si APD. The incident photons are mainly absorbed at the photon absorbing region where most electron-hole pairs are generated. These electrons are then swept to the multiplication region where the multiplication of carriers occurs. The electrical field inside photons absorbing region and carrier multiplication region is controlled by the doping profile of the charge layer, which is the key to the Si APD design. The photon absorbing region can be fully depleted by a high-enough electrical field before significant carrier multiplication occurs at the multiplication region. Thus, the classic reach-through structure is utilized. In addition, generally, the photons absorption and the multiplication of carriers take place at different regions, which are called the separation absorption, charge, and multiplication (SACM). Figure 1b is the optical microscopy image the fabricated device. The photosensitive area (the circular area in earthy gray color) is surrounded by the anode metal, which can act as baffle for stray light in practical applications.

The designed Si APD is fabricated using 0.5 μm CMOS technology without any process modification. The 8-inch p-type <100> wafer used to fabricate the device comprises a 12 μm-thick silicon epitaxial layer with resistivity of 1000 Ω·cm and a thick substrate with resistivity of 0.01 Ω·cm. Figure 2 shows the key process for the fabricated Si APD. First, the wafer was implanted with boron and phosphorus in sequence to form the p-tub region (p-well) and n-type guard-ring (n-well) as shown in Figure 2a. Second, field oxide (FOX) is used to isolate the regions with different type doping and the adjacent devices, which was implemented by the local oxidation of silicon (LOCOS) technology, as shown in Figure 2b. At the same time, the drive-in for p-well and n-well is accomplished by the high temperature (>1000 ℃) process used in LOCOS. Third, the charge layer was formed by an implantation with high-energy and low-dose followed by a high temperature annealing in furnace, as shown in Figure 2c. The doping profile of the charge layer plays a determinative role in controlling the electric field inside the photons absorbing region and the multiplication region. However, the implanting energy is limited in a modern foundry so that the peak doping concentration position (projected range, $R_{p}$) of the charge layer is consequently limited. Fortunately, the $R_{p}$ can be increased by the channeling effect, which should be avoided in most cases [18]. Thus, the wafer cleaning before charge layer implantation is very important. Fourth, the heavy doping for Ohm contacts was formed by sequentially implanting the patterned region with boron and phosphorus in low energy and high dose, as shown in Figure 2d. A high-temperature annealing was conducted immediately to activate the dopants and repair the implanting damage. Fifth, the tungsten plus were formed to reduce the connection resistivity after the chemical-mechanical polishing (CMP) of the intermetallic dielectric layers 0 (ILD0), as shown in Figure 2e. Finally, the AlCu alloy was deposited and patterned to form the metal pads, as shown in Figure 2f. 

## 3. Device Characterization and Analysis

### 3.1. Dark Current and Photocurrent

The dark and photocurrent of the Si APD are depicted in Figure 3. The dark current is measured under a no-illumination environment, which is implement by an electromagnetic shielding box. The accurate track of the voltage and current is accomplished with a Keysight B2902 precision source/measure unit. Moreover, the voltage step is reduced from 0.5 V to 0.1 V near the breakdown point to accurately capture the current change. The current compliance is set to 1 mA to prevent the device under test (DUT) from permanent damage.

Generally, the dark current of the DUT is far below 1 nA when the applied bias is lower than 140 V. Such low-level dark current is comparable to some commercial products such as the C30902SH-2 series Si APD from Excelitas and the S14645-02 series from Hamamatsu. The abnormal trend in dark current at low bias (<10 V) may be attributed to the system leakage and the discharge of the long wires between the probes and the output connectors of the measurement unit. The dark current increases at a slow rate when the applied reverse voltage (*V_r_*) is lower than the 145 V due to the weak multiplication. However, as *V_r_* further increases, the dark current increases very quickly because of the pronounced avalanche effect. It reaches 100 μA at 148, V which is defined as the breakdown voltage (*V_BD_*) for the fabricated Si APD.

A single-mode semiconductor laser with wavelength of 905 nm is used to characterize the photocurrent of the Si APD. The output optical power is approximately −29.8 dBm, which is checked by an optical power meter PM20 from Thorlabs. The incident optical power is estimated to be −30.8 dBm, specifically 832 nW, considering about 1 dB loss from the fiber and reflection. The photocurrent initially increased in a relatively gentle trend until *V_r_* reaches 146 V due to the relatively weak multiplication effect. The photocurrent increases at a slower rate as *V_r_* reaches the punch-through voltage (*V_rt_*) at which the photon absorbing region is fully depleted. Such change is demonstrated in Figure 3b. With remarkable avalanche multiplication, the photocurrent rises very rapidly and reaches a plateau caused by the current compliance as shown in Figure 3c.

High temperature stability of the breakdown voltage is vital to the avalanche photodetector because of close dependency of characteristics on temperature. Therefore, the dark current measurement under ambient temperature ranging from 233 K to 393 K was performed, and the result is shown in Figure 4a. The dark current curves at 233 K and 273 K almost coincide before the *V_r_* reaches the breakdown point. This is because the measurement system leakage is at the level of 1~10 pA. It is clear that the dark current under the same bias increases with the temperature due to the enhanced thermal excitation at elevated temperature [19]. Another remarkable feature in Figure 4 is that the *V_BD_* increases with the temperature, which can be attributed to the cooling effect caused by the enhanced phonons scattering at high temperature [20]. The dependence of the breakdown voltage on ambient temperature is depicted in Figure 4b. Linear fitting is then performed to extract the temperature coefficient of the breakdown voltage. The temperature coefficient of the fabricated device is estimated at 0.077 V/K. Such low temperature coefficients can be attributed to the limited width of the multiplication region. The ion energy used in ion implantation for the charge layer limits the project range of the doping profile, which is approximately the width of the multiplication region. According to [21,22], the phonon scattering effect on the carriers can weaken as the multiplication region is narrowed. Consequently, the dependence of breakdown voltage on temperature is also weakened. 

### 3.2. Terminal Capacitance

Figure 5 shows the measured terminal capacitance of the device under different bias. The voltage step is 1 V and the measurement frequency is 1 MHz. It can be clearly seen that the capacitance sharply drops to 1.6 pF at 16 V, which is defined as the punch-through voltage. This is completely consistent with the inflection point in the photocurrent in Figure 3. The 12 μm-thick epi-Si layer is completely depleted at punch-through voltage, which is considered as the unit gain point. The terminal capacitance of the fabricated device is slightly higher than that of some Si APD products from Hamamatsu and First Sensor because of the much narrower epi-Si layer. 

At the same time, the punch-through voltage can be estimated by the following equation [23]:(1)$$V_{\mathrm{rt}}=\frac{{\mathrm{qQ}}_{p}w_{a}}{\varepsilon_{\mathrm{Si}}\varepsilon_{0}}$$
where *q* is the elementary charge, *Q_p_* is the number of uncompensated charges per unit area contained in the charge layer and is nearly same as the dose of implanted boron, *w_a_* is the width of the multiplication region, which is almost equal to the project range of boron implantation, and *ε**_Si_* and *ε*_0_ are the relative dielectric constant of silicon and the permittivity in vacuum, respectively. For the fabricated device, *Q_p_* is $1\times 10^{12}\;{\mathrm{cm}}^{-2}$, and *w_a_* is about 1 μm, which is verified by secondary ion mass spectrometry (SIMS). Substituting the above data into Equation (1), *V_rt_* is approximately 15.2 V. The discrepancy between the calculation and the measurement may be attributed to the fact that the punch-through voltage consists of two parts. The main part results from the depletion of the charge layer. Another smaller part is introduced by the depletion of the photon absorption region. The voltage-drop caused by depleting the charge layer is much higher than that used to deplete the photo absorption region. This is because of three orders higher doping level inside the charge layer than that of the Epi-Si layer. 

### 3.3. Responsivity and Gain

The voltage dependence of responsivity is shown in Figure 6a. The responsivity is estimated through the equation,
(2)$$R(V)=\frac{I_{\mathrm{net}}\left(V\right)}{P_{\mathrm{in}}}=\frac{I_{\mathrm{ph}}\left(V\right)-I_{d}\left(V\right)}{P_{\mathrm{in}}}$$
where *I_net_**(V)* is the net photocurrent and is equal to the difference between the total measured photocurrent *I_ph_(V)* and the dark current *I_d_(V)*; *P_in_* is the total optical power entering the photodetector. The primary responsivity, which is defined as the responsivity at *V_rt_* is estimated to 0.22 A/W. The primary responsivity is lower than that of some commercial products because of the narrower photon absorbing region. The responsivity grows slowly until the reverse bias reaches 146 V. This can be attributed to the weak avalanche multiplication effect corresponding to not-high-enough electric field inside the multiplication region. The electric field inside the multiplication region is determined by the doping profile of the charge layer, especially the peak concentration and peak position. The pronounced avalanche effect under elevated bias results in a faster rising trend in responsivity. The responsivity reaches the maximum, 1006.5 A/W, at the breakdown voltage and is followed by a sharp decline due to the self-quenching effect [24], as clearly shown in Figure 6b.

The net current gain can be extracted from the I-V curve and the R-V curve through the equation
(3)$$G(V)=\frac{I_{\mathrm{ph}}\left(V\right)-I_{d}\left(V\right)}{I_{\mathrm{ph}}{(V}_{\mathrm{rt}})-I_{d}{(V}_{\mathrm{rt}})}=\frac{R(V)}{{R(V}_{\mathrm{rt}})}$$
where *I_ph_*(*V_rt_*) and *I_d_*(*V_rt_*) are the photocurrent and dark current at the punch-through voltage, and *R*(*V_rt_*)is the primary responsivity. It is clearly shown in Figure 6c that the growth trend of the current gain is almost same as the responsivity due to the tight relationship between them, as shown in Equation (3). The maximum current gain, 4610, was also measured at the breakdown voltage, as depicted in Figure 6d. The voltage at which the gain reaches 100 is 146.4 V for the fabricated device, which is the operating point for most commercial products.

## 4. Conclusions

In this work, a silicon avalanche photodetector with SACM configuration is designed and implemented with 0.5 μm CMOS technology without any process modification. The fabricated device shows a low-level dark current, low breakdown voltage, ultra-low temperature coefficient of the breakdown voltage of 0.077 V/K, and high current gain. The strong photoresponse of the device at near infrared wavelength provides a very promising solution for light detection for near-infrared LiDAR applications.

Limited by the existing test equipment, the spectral response, the response speed, and the multiplication noise are not able to be presented in this paper. These characteristics are the focus of future research. Although the very high responsivity and multiplication gain are demonstrated at near-infrared wavelength, the corresponding bias is the breakdown voltage at which the dark current cannot be ignored. Furthermore, the low primary responsivity of the fabricated device can be improved by increasing the thickness of the epitaxial silicon layer. The operating voltage window for practical application is narrow because of the limited width of the multiplication region, and can be widen by increasing the implantation energy for the charge layer.

## Figures and Tables

**Figure 1 micromachines-13-00047-f001:**
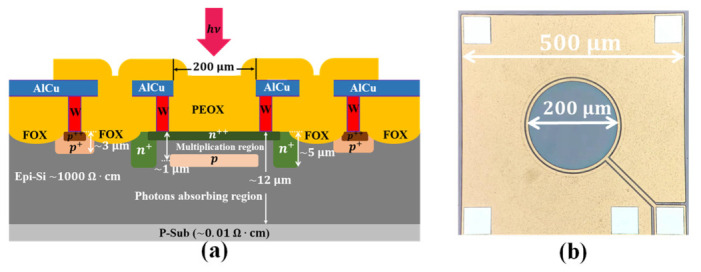
Structure of the silicon avalanche photodetector (**a**) cross-section schematic; (**b**) optical microscopy image.

**Figure 2 micromachines-13-00047-f002:**
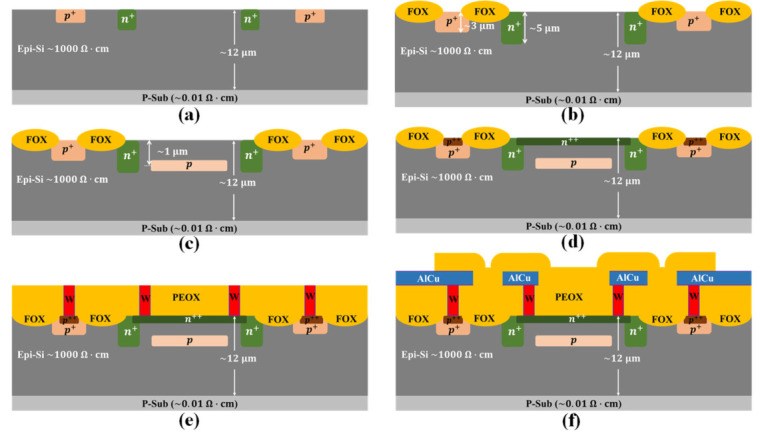
Key fabrication process of the Si APD: (**a**) implantation of P-well and N-well; (**b**) LOCOS and drive-in for P-well and N-well; (**c**) implantation for charge layer and annealing; (**d**) implantation for Ohm-contacts and rapid thermal annealing; (**e**) ILD0 deposition, CMP and tungsten-plugs formation; (**f**) AlCu deposition, patterning, and passivation.

**Figure 3 micromachines-13-00047-f003:**
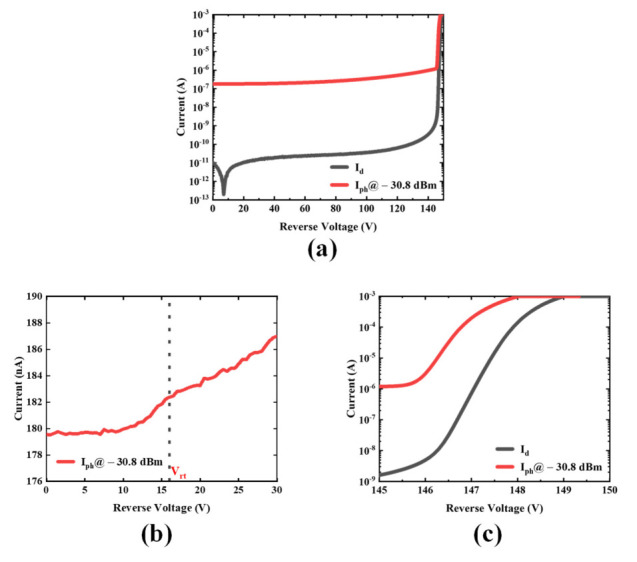
Current versus reverse voltage of the device: (**a**) the current versus reverse voltage from 0 V to 150 V; (**b**) the photocurrent under low bias voltage; (**c**) the dark current and photocurrent near the breakdown point.

**Figure 4 micromachines-13-00047-f004:**
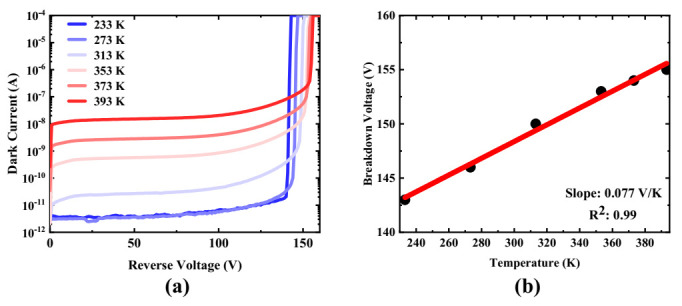
Temperature characteristics of the device: (**a**) dark current versus reverse voltage under different temperature; (**b**) breakdown voltage versus temperature (black dot: measured data, red line: linear fit).

**Figure 5 micromachines-13-00047-f005:**
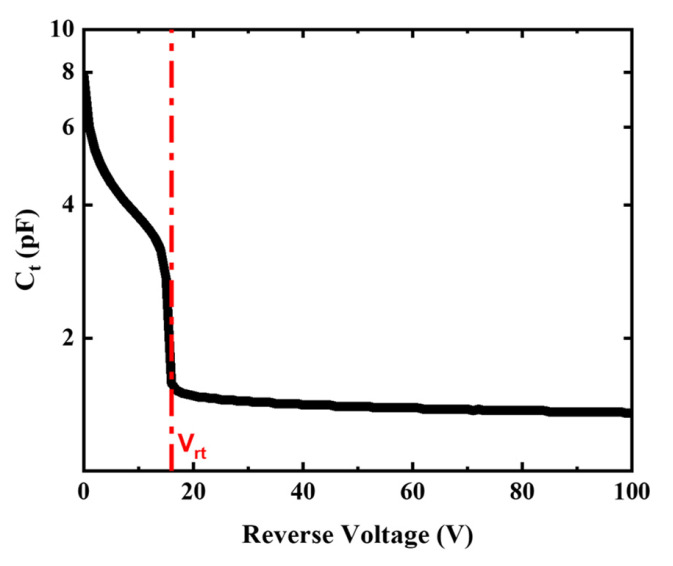
Capacitance versus reverse voltage of the Si APD.

**Figure 6 micromachines-13-00047-f006:**
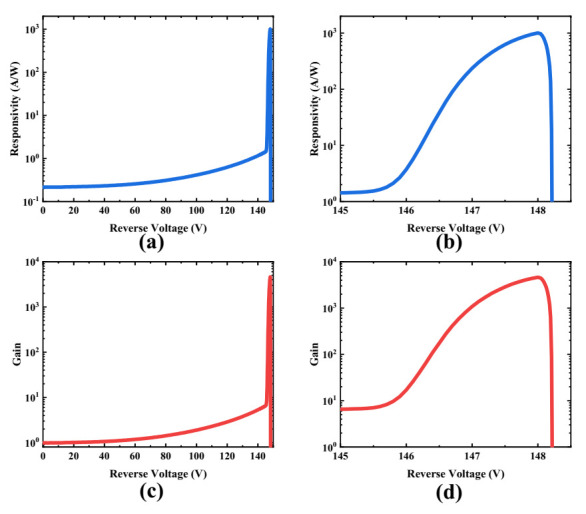
Static photoresponse of the Si APD: (**a**) responsivity versus reverse voltage; (**b**) responsivity near breakdown voltage; (**c**) net current gain versus reverse voltage; (**d**) net current gain near breakdown voltage.

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
