# Peer review of "A Near-Infrared CMOS Silicon Avalanche Photodetector with Ultra-Low Temperature Coefficient of Breakdown Voltage"

_micromachines, 2021, doi:10.3390/mi13010047_

Round 1

Reviewer 1 Report

The submitted paper is a nice presentation of interresting experimental results.

I have only several comments for authors:

1) The title of paper and generally the interpretation of expected spectral dependences.
The most of properties of Silicon are quite flat or monotone in wide spectral region. I am understand that all measurements have been completed using 905 nm laser only and this one wavelength is interesting from point of view of forseen application. But according my knowledge there is no reason to enhance this value by this way with this accuracy in the title and also in the conclusion. The response of your structure at 900 nm will be practically the same and at 800 nm very similar. I am understand that authors would like to distinguish from the green and blue wavelength region, but I suggesting do it by other way than exact value 905 nm.

2) The classification of device on compare with others.
Starting read the introduction it was little bit unclear to understand what kind of device is described by authors. Well, the classification (p1 starting row 36) is nice, but the presented kind of device is usually called "linear APD" or "linear operated APD" to enhance its difference from non-linear Geiger mode of operation of single-photon APD. It should be added in Introduction.

3) Figure insets.
From point of view of physics of photodection, the insets in Fig. 3 and Fig. 6 are more interresting than main plots. And they are referenced in text. I sugest invert their sizes or make all figures the same size.

Reviewer 2 Report

The manuscript presents a silicon APD with punch-through structure with low level dark current and high gain. The comments are as follows:

  1. It is nonsense to say " to our best knowledge". Such sentence should be deleted to avoid any possible confusion.
  2. Some recent published papers on APD should be included in the references. It seems the authors didn't give a thorough review on the related topic. Several datasheet files are listed in the references currently. It is really rare in scientific publications. The paragraph between L44-56 seems to report the products instead of the technique progress. I don't think it is suitable to be published in the academic journal.
  3. The data of relevant doping concentrations are missed. Please supplement the key information. Also, the depth of each active region should be labelled in Fig.1 and Fig.2. The authors said the 0.5 um CMOS technology. It is not a common-used technology currently. 
  4. The size of the cell in Fig.1(b) should be included.
  5. The equation in L188 should be labelled as Eqn(1).
  6. The sentence in L195-197 is not scientific. The reason of the discrepancy should be addressed in detail.

Round 2

Reviewer 2 Report

The uploaded revised manuscript looks abnormal. The files should be the clear version.

Round 3

Reviewer 2 Report

The authors have answered my questions. I don't have any concerns on the manuscript anymore.